# Influence of Mine Environmental Factors on the Liquid CO_2_ Pipeline Transport System with Great Altitude Difference

**DOI:** 10.3390/ijerph192214795

**Published:** 2022-11-10

**Authors:** Guansheng Qi, Hao Hu, Wei Lu, Lulu Sun, Xiangming Hu, Yuntao Liang, Wei Wang

**Affiliations:** 1College of Safety and Environmental Engineering, Shandong University of Science and Technology, Qingdao 266590, China; 2State Key Laboratory of Strata Intelligent Control and Green Mining Co-Founded by Shandong Province and the Ministry of Science and Technology, Shandong University of Science and Technology, Qingdao 266590, China; 3College of Safety Science and Engineering, Anhui University of Science and Technology, Huainan 232063, China; 4State Key Laboratory of Coal Mine Safety Technology, China Coal Technology & Engineering Group, Shenyang Research Institute, Shenyang 113122, China

**Keywords:** liquid CO_2_, pipeline transportation, environmental factors, great altitude difference, coal mine

## Abstract

To prevent coal spontaneous combustion and store CO_2_ in the coal mine, it is necessary to establish a fire-prevention pipeline transport system which continuously injects a large amount of liquid CO_2_ from the ground to the underground area directly. At present, few studies are focused on the law of liquid CO_2_ transport with great altitude difference. Moreover, the complex transport environment in the coal mine affects the design and application of the pipeline transport system for ground direct injection of liquid CO_2_. This study explores the influence of environmental factors at different depths in the coal mine on the liquid CO_2_ transport. Excessive altitude difference, ambient temperature and airflow velocity may lead to the boiling of liquid CO_2_ during pipeline transport and a sudden drop in CO_2_ temperature and pressure, which may cause danger in the pipeline transport system. The critical insulation thickness is determined based on the occurrence of the boiling of CO_2_. In addition, the influence law of adding an insulating layer of different thicknesses to the CO_2_ pipeline system is obtained. This study is of great significance to the establishment of a pipeline system that safely transports liquid CO_2_ from the ground to the underground mine.

## 1. Introduction

Coal spontaneous combustion (CSC) is one of the major disasters in coal mines [1,2]. The exothermic reaction between the broken coal and oxygen in the gob makes the rate of heat generation higher than that of heat dissipation. As a result, CSC occurs when coal heats up and reaches the ignition point [3,4,5]. Considering factors such as the complex geological conditions and wide space, it is difficult to prevent and extinguish fires caused by CSC in the gob [6]. The injection of liquid carbon dioxide (CO_2_) into the gob serves as a good method to prevent CSC [7,8,9]. When liquid CO_2_ is injected into the gob, it gasifies rapidly and absorbs a lot of heat, which greatly reduces the overall temperature of the gob and inhibits heat accumulation [10]. Moreover, gasified CO_2_ is of a higher density than the air, so it adheres to the coal seams in the gob to hinder coal oxidation. In doing so, CSC is effectively prevented [11].

In addition, direct injection of liquid CO_2_ to prevent fire can lower the emission amount of greenhouse gas. Carbon capture and storage (CCS), a potential and scalable technology, can reduce the shares of emission of electricity generation technologies by 19% from 2010 to 2050 [12,13]. CO_2_ can be transported from its source (such as coal chemical plants, power plants, etc.) to the injection point by tank truck, ship, pipeline, etc. [14,15]. In addition, CO_2_ can be injected into oil reservoirs for enhanced oil recovery (EOR) [16,17,18], unrecoverable coal seams for enhanced coal bed methane (ECBM) recovery [19,20] and saline aquifers [21,22,23]. Hence, the injection of CO_2_ into abandoned gobs for storage has promising prospects. At present, in China alone, there are about 7000 coal mines and plenty of gobs where CO_2_ can be injected. Furthermore, studies have found that some industrial wastes (such as coal gangue and carbide slag) can be used to further permanently solidify CO_2_ [24,25,26], and this technology has been applied in practical engineering.

To meet the requirements of underground fire prevention and large-scale CO_2_ storage in the gob, it is necessary to establish a pipeline transport system that can inject liquid CO_2_ from the ground to the underground area.

Nowadays, methods of CO_2_ transported by pipeline in mine fall into two types: gaseous CO_2_ transport and liquid CO_2_ transport [9] because CO_2_ in solid state or under high temperature and pressure is not suitable for the fire prevention system considering safety and cost [27,28,29]. Since gaseous CO_2_ transport is of a low density and high pressure drop, it requires pipelines of a much larger diameter than other transport methods. Moreover, it may need more energy to offset the loss caused by the pressure drop [30]. In contrast, liquid CO_2_ can avoid the loss. Generally, liquid CO_2_ can be transported to the underground area in the following two ways. One is to drill holes above the gob and then inject CO_2_ into the gob through pipelines, yet it costs a lot, and CO_2_ can only be injected into one gob, displaying a lack of flexibility [31]. The other way is to inject liquid CO_2_ into the storage tank on the ground and then transport the tank to the designated underground location for fire prevention, which improves flexibility, but CO_2_ cannot be continuously injected in a huge amount [32]. Therefore, it is necessary to establish a pipeline system that can continuously inject quantities of liquid CO_2_ from the ground to the gob.

At present, in most cases, CO_2_ is captured from its gas source and then transported in a dense phase (liquid or supercritical phase) in a large scale to the injection points, such as oil reservoirs and saline aquifers [23,33]. Zhang et al. analyzed the pressure distribution of subcooled CO_2_ in liquid and supercritical phase during the transport under isothermal and adiabatic conditions, respectively. They also studied the maximum transport distance at different inlet temperatures [30]. Witkowski et al. studied the temperature and pressure variations along the CO_2_ transport path under different ambient and inlet fluid temperatures based on two state equations, namely LKP and PRBM [34]. Teh et al. concluded the influence law of CO_2_ on the on-way transport parameters through buried and non-buried transport in cold and thermal environments [35]. This study is focused on the influence of environmental factors on CO_2_ transport.

Coal mines correspond to a complex environment. There is a great altitude difference (up to 1000 m) from the CO_2_ cylinder on the ground to the gob for the liquid CO_2_ pipelines. Furthermore, it is difficult to control the on-way parameters of liquid CO_2_ in the pipeline by installing equipment in the vertical transport pipeline in boreholes or roadways. The liquid CO_2_ in the pipeline causes great pressure due to the great altitude difference. Moreover, CO_2_ goes through a long roadway before it enters the gob. Terrestrial heat can cause an extremely high temperature in the deep coal mine; and the ventilation requirements result in a high airflow velocity in some roadways. These two factors then lead to a strong heat convection between pipelines and the ambient environment. Hence, in the CO_2_ transport system for fire prevention, the temperature and pressure of CO_2_ should be strictly controlled to guarantee the liquid phase of CO_2_; otherwise, the boiling of CO_2_ may cause damage such as ice blocking and vibration of pipelines [18,36], which will affect the operation of pipeline transport system.

Heat conduction is a type of heat transfer caused by the temperature difference inside or between objects, which does not involve mass flow mixing. The heat transfer efficiency is determined by the heat conductivity of an object [37]. Materials with appropriate thermal insulation are widely used in buildings and industrial fields to control the heat loss/heat gain between surfaces in operation above/below ambient temperatures by reducing the heat conductivity between surfaces [38]. In general, pipelines that are not actively heated or cooled may require thermal insulation measures to reduce temperature variation. Kayfeci found that the addition of an insulating layer to the pipeline can ensure the economic efficiency of the transport system [39]. The temperature of liquid CO_2_ is generally lower than the ambient temperature during the transport in the coal mine. Therefore, it is necessary to add an insulating layer in the transport system to slow down the heat-absorption of the fluid in the pipeline from the ambient environment.

In this study, the CO_2_ transport process in the fire prevention system is simulated based on the commercial simulation software ASPEN HYSYS V8.4^®^. Firstly, on-way parameters are obtained, such as the temperature and pressure along the pipelines without insulating layer at different depths and under different thermodynamic conditions. Then, the two-phase flow under different conditions is studied to determine whether an insulating layer of a certain thickness can change the two-phase flow into a pure liquid flow. If the insulating layer works, the optimum insulation thickness of the whole transport process can be determined by dichotomy, which provides a basis for designing the fire prevention system of direct liquid CO_2_ injection.

## 2. Methods

### 2.1. CO_2_ Pipeline Transport (Direct Injection) System in Coal Mine

The pipeline transport (direct injection) system includes a storage tank, CO_2_ transport pipelines, valves, flowmeters and different kinds of sensors (such as a temperature sensor) (Figure 1). The storage tank is usually placed near the upcast shaft, which minimizes heat transfer on the ground and reduces the pipeline cost. Therefore, the calculation of the ground pipeline is negligible in the study. The upcast shaft is in the ventilation system to discharge polluted air. A strong fan in the mine makes the air temperature in the return airway close to that in the underground tunnel. The strong fan is set at the end of the upcast shaft to discharge underground polluted air and heat from the shaft, so that the upcast shaft is of a uniform temperature, i.e., the ambient temperature of the upcast shaft and the temperature of the underground roadway are the same. The vertical transport pipelines are installed in the upcast shaft, so the pressure of the fluid in this pipeline is affected by gravity due to the altitude difference; both vertical and underground horizontal pipelines are affected by high airflow velocities and high temperature in coal mines. In this study, influencing factors such as the coal mine depth and thermodynamics factors (airflow velocity and ambient temperature) were studied.

### 2.2. Research Methods

Commercial simulation software ASPEN HYSYS V8.4^®^ is widely adopted to simulate the CO_2_ pipeline transport process [35,40,41]. Luo et al. used this software to conduct a technology-economy assessment of CO_2_ transport pipelines in Humber, UK. Then they compared and contrasted the calculation results with those of PIPEFLO and found the results were of highly consistency [41].

CO_2_ transported in pipelines requires an accurate calculation of the physical properties of CO_2_, such as phase behavior, density, viscosity, etc. The Peng–Robinson (PR) equation of state (EOS) has been widely used to calculate the thermodynamic properties and vapor liquid equilibrium (VLE) of CO_2_ in engineering [28,29,40,42]. Veritas held that the PR equation can predict the mass density of liquid CO_2_ [43]. Li et al. found that the PR equation was of high precision for calculating the physical properties of pure CO_2_ or CO_2_ with impurity like sulfuretted hydrogen or methane [44]. After CO_2_ is injected into the simulated environment, the inlet parameters and pipeline parameters during CO_2_ transport were set (Figure 2). The inlet parameters include the CO_2_ flow, inlet temperature and inlet pressure. There is a certain relationship between the CO_2_ flow and the transport distance. When other parameters (such as inlet temperature, pipe diameter, etc.) are unchanged, the faster the CO_2_ flow is, the more notably the pressure drops during CO_2_ transport, and the shorter the transport distance is. The simulation adopted a fast flow of 5000 kg/h, which was in the demand flow range for CO_2_ fire prevention [9,45]. Inlet temperature and pressure also play an important role in the transport process. The transport distance can be adjusted through the variations of temperature and pressure. In the CO_2_ transport system, the CO_2_ source is a fixed storage tank or a tank truck on the ground. In general, the inlet temperature (i.e., outlet temperature of the storage tank) was no higher than −20 °C, and the pressure was in the range of 4–24 bar. The inlet temperature and pressure during the simulation were set as −20 °C and 22 bar, respectively [9,11].

The setting of the pipeline parameters includes basic characteristic parameters, such as pipeline diameter, wall thickness, pipeline length and altitude difference, and thermodynamics-related parameters, such as the type and properties of the medium around the pipeline and the insulating layer. In this study, the pipeline is made of mild steel, whose Darcy–Weisbach roughness is 4.572 × 10^−5^ m [46]. The pipeline diameter was calculated based on the flow velocity [33,47,48]:(1)$$\mathrm{ID}=\sqrt{\frac{4m}{\upsilon \pi \rho }}$$
where m is the mass flow rate (m^3^/s); υ represents the flow velocity (m/s); ID is the inner diameter (m); ρ is the CO_2_ density. The inner diameter of the pipelines is 29.3 mm, and the standard value of pipe diameter of 32 mm is adopted [49]. The wall thickness is calculated by [50,51]:(2)$$\delta_{1}=\frac{p_{\mathrm{re}}D_{\mathrm{in}}}{2\sigma_{s}F\varphi }$$
(3)$$\delta =\delta_{1}+C_{1}+C_{2}$$
where δ is the designed wall thickness of the pipeline; δ_1_ is the theoretical wall thickness; pre is the maximum operating pressure of the pipeline (Pa) and is set as 16 MPa; Din is the outside pipeline diameter (m); $\sigma_{s}$ is the specified yield stress for the pipe material (Pa). In this study, Q345E steel pipe is selected, so $\sigma_{s}$ is 345 MPa. F represents the longitudinal joint factor, set as 1 for seamless steel pipe; φ is the design factor and is set as 0.72; C_1_ is the corrosion allowance, and its value is in the range of 1–2 mm; C_2_ is the added value of the wall thickness deviation of the steel pipe, and its value is 0.15 times of the value of δ_1_.

Considering the great pressure caused by the great altitude difference of the vertical pipeline, the pressure of the pipeline is set as 16 MPa, the yield strength of the steel as 345 MPa (GB/T1591-2018, standard for low-alloy and high-strength structural steel), the welding coefficient as 1 (seamless steel pipeline) and the corrosion allowance as 2 mm. In addition, the wall thickness is rounded up to 4 mm. Pipelines in the simulation fall into two sections, i.e., the vertical pipeline section in the return airway and the underground horizontal pipeline section. Factors such as elbows, valves, etc., are negligible in the simulation. The huge altitude difference in the vertical pipeline exerts a great influence on CO_2_ transport. At present, the depth of the coal mine is generally no more than 1000 m, so 200 m, 400 m, 600 m, 800 m and 1000 m were selected as the depths of vertical pipelines in this study to explore the liquid CO_2_ transport. The underground pipeline is 2000 m long horizontally.

Thermodynamic parameters are set to calculate the heat transfer coefficient of the pipeline. The modes of heat transfer mainly include heat conduction, radiation and convection. Since the transport pipelines receive little thermal radiation, the simulation just explores the effects of heat conduction and heat convection. In general, the ambient temperature is higher than that of the liquid CO_2_ in pipelines. According to the second law of thermodynamics, the liquid CO_2_ in the pipeline spontaneously absorbs heat from the ambient environment, which leads to a temperature rise during transport. This process includes three steps. Firstly, the ambient air exchanges heat with the outer wall of the pipeline (or the outer wall of the insulating layer) through heat convection. Secondly, heat is transferred between the outer wall and inner wall of the pipeline (or the outer wall of the insulating layer) through heat conduction. Thirdly, heat exchange occurs between the internal fluid and the surrounding pipeline wall through heat convection. Based on the law of conservation of energy, the above three correspond to an equal amount of heat transfer. Heat convection is related to Prandtl number and Reynolds number, q = h × (T_w_ − T_∞_), where q is the heat exchanged between the solid surface per unit area and the fluid in unit time, which is called heat flux (W/m^2^); Tw and T∞ are the temperatures of the solid surface and fluid, respectively (K); Q = h × A × (T_w_ − T_∞_) = q × A, where A is the wall surface area (m^2^); Q is the heat transfer over area A per unit time (W); h is the heat transfer coefficient of surface convection (W/(m^2^·K)), which is determined by both Prandtl number and Reynolds number. The heat transfer is calculated based on Fourier’s law:(4)$$q=-k\frac{\mathrm{dT}}{\mathrm{dx}}$$

Firstly, on-way parameters of CO_2_ during the transport at different depths and different ambient temperatures without insulating layer were studied. The ambient temperatures in the study were 20 °C, 25 °C, 30 °C, 35 °C and 40 °C, respectively. Furthermore, air flow is crucial to heat convection and was discussed at 1 m/s, 7.5 m/s and 15 m/s (according to Article 136 of Coal Mine Safety Regulations [52]). Secondly, experiments were conducted to explore whether the addition of an insulating layer can prevent two-phase flow in the pipeline. If it works, the insulation thickness, with which two-phase flow does not occur, can be obtained. The thermal conductivity of the insulation is set as 0.040 w/m·k. This process can be obtained by dichotomy as follows.

Step 1. Set the initial thickness as h_0_ = 1 m and recalculate it when two-phase flow occurs. If there is no two-phase flow in the entire pipeline, then set the thickness as h_1_ = (h_0_ + 0)/2. Otherwise, it can be concluded that it is not economically beneficial to use an insulating layer.

Step 2. If no two-phase flow occurs in the entire pipeline; then h_2_ = (h_1_ + 0)/2. Otherwise, h_2′_ = (h_0_ + h_1_)/2.

Step 3. Repeat Step 2 until the precision of h_n_ is 0.0001 m.

### 2.3. Numerical Model Validation

The accuracy of the model is verified by the experimental data of liquid CO_2_ direct injection in II020210 fully mechanized top coal caving face of Yangchangwan Coal Mine [53]. The direct injection pipeline transport system includes tank cars, 460 m vertical drilling pipelines, 700 m underground pipelines and various valves. The main measuring points include ground tank cars, underground chambers and underground pipeline outlets. Table 1 gives the data of temperature and pressure.

The simulation results are given in Figure 3.

It can be seen that the outlet pressure of the pipeline is 18 bar, in the range of 15–20 bar of the test results; and the outlet temperature of the pipeline is −25.3 °C, in the range of −30 °C to −20 °C, which is also in line with the test results. Therefore, the calculation results are consistent with the actual situation.

## 3. Results and Discussions

### 3.1. Variation of CO_2_ On-Way Parameters with Depth under Different Conditions

Three factors can influence CO_2_ on-way parameters, namely, the depth of the vertical pipeline, ambient temperature and airflow velocity.

Figure 4 and Figure 5 show the on-way variations of temperature and pressure along a 1000 m vertical pipeline. According to Figure 4a, at 1 m/s airflow velocity, temperature curves are almost straight lines because temperatures rise at constant rates in the 1000 m pipeline. When the ambient temperatures are 20 °C, 25 °C, 30 °C, 35 °C and 40 °C, respectively, the outlet temperatures of the vertical pipeline are 1.55 °C, 3.54 °C, 5.49 °C, 7.41 °C and 9.29 °C, respectively. A higher ambient temperature causes a higher outlet temperature of the pipeline. The reason is that the heat transfer rate is subject to the temperature difference between the ambient temperature and the inlet temperature when the surrounding wall material and airflow velocity remain unchanged.

In Figure 4a–c, at 40 °C ambient temperature, airflow velocities of 1 m/s, 7.5 m/s and 15 m/s correspond to outlet temperatures of 9.29 °C, 28.04 °C and 32.46 °C, respectively. A higher airflow velocity leads to an increase in the outlet temperature of the pipeline; and it can be seen that the difference of the outlet temperatures at 1 m/s and 7.5 m/s airflow velocities is greater than that at 7.5 m/s and 15 m/s, which indicates that airflow velocity exerts a greater influence in the range of 1–7.5 m/s than in 7.5–15 m/s. This is because the outlet temperature at 7.5 m/s airflow velocity is quite close to the ambient temperature, thus weakening the effect of the overall heat exchange on the on-way temperature.

A contrastive analysis on the pressure curves (Figure 5) finds that the higher the ambient temperature is, the slower the pressure rises. When the airflow velocity is 1 m/s (Figure 5a), the pressure curves basically coincide with each other. However, a higher airflow velocity results in a more notable separation of pressure curves along the pipeline, which illustrates that compared with the airflow velocity (Figure 5b,c), the ambient temperature has a less impact on the variation of the pressure gradient of the vertical pipeline. The reason is that a higher airflow velocity corresponds to a faster temperature rise, resulting in a decrease in the liquid CO_2_ density and an increase in the volume flow rate. Resultantly, the flow velocity becomes higher, and the pressure drops more notably. According to Figure 4 and Figure 5, under different conditions, although the fluid temperature in the vertical pipeline gradually rises, even up to 32.46 °C, no two-phase flow occurs in the entire vertical pipeline. The reason is that the pressure rises with the depth due to the gravity of the fluid in the vertical pipeline. For pipelines of 200 m, 400 m, 600 m and 800 m depths, their temperature and pressure variations are calculated in the same way, so their results are consistent with those in Figure 4 and Figure 5. For example, the result of a 200 m pipeline is consistent with that of the first 200 m part of the 1000 m pipeline in Figure 4 and Figure 5. Hence, it can be concluded that a shorter depth leads to similar values of outlet temperatures and pressures.

In addition, the temperature and pressure at the end (outlet) of the vertical pipeline are regarded as the inlet temperature and pressure of the underground horizontal pipeline. When the ambient temperature and the airflow velocity rise, the inlet temperature rises, yet the pressure of the underground pipeline falls.

Figure 6, Figure 7, Figure 8, Figure 9, Figure 10 and Figure 11 display the variations of on-way temperature and pressure during CO_2_ pipeline transport (both vertically and horizontally) under the conditions of different airflow velocities, depths and ambient temperatures. In general, both on-way temperature and pressure have one or two characteristic points (continuous but non-differentiable points). When CO_2_ is transported to the intersection of the vertical pipeline and the horizontal pipeline, the first characteristic point occurs: the on-way temperature rises at a falling rate and the on-way pressure reaches its maximum and begins to decline. The second characteristic point is in the horizontal pipeline section under certain conditions. After the second point, the on-way temperature and pressure plunge, indicating the gasification (boiling) of CO_2_.

The temperature-rise rates before and after the first characteristic point differ, which suggests different heat transfer rates. Since the ambient temperature, airflow velocity, wall thickness and roughness of the vertical pipeline are the same, the difference is caused by the pressure variation due to the gravity of the fluid. Thus, the heat transfer caused by the ambient temperature will lead to a temperature rise of the fluid, thus reducing the density of the fluid. Pressure increases with the depth of vertical pipelines (Figure 9, Figure 10 and Figure 11), which offsets the decrease in density caused by the temperature rise and may even lead to an increase in density. However, in the horizontal pipeline, the pressure gradually decreases due to the friction with the pipeline wall. The increase in density accelerates the heat transfer, which explains the first characteristic point.

The second characteristic point is the starting point of CO_2_ gasification. When CO_2_ in the pipeline is transported from the liquid phase area to the boundary line of gas–liquid phase area, it gasifies gradually. Meanwhile, the increase in gasified CO_2_ with a low density leads to a decrease in the average density. Thus, the volume flow rate increases, which raises the flow rate. As a result, the pressure gradually drops. This explains the gradual decrease in pressure after the second characteristic point. Gasification lasts for a long time. In a three-phase diagram, the temperature and pressure point vary along the dew-point curve toward a lower pressure (energy). Hence, temperature declines as the pressure gradually decreases.

According to Figure 6, Figure 7, Figure 8, Figure 9, Figure 10 and Figure 11, in most cases, the gasification of CO_2_ in the pipeline occurs at different positions. When the ambient temperature serves as the only variable, gasification is more likely to occur at a higher ambient temperature. For example, in Figure 6c, for a 600 m deep coal mine with ambient airflow velocity of 1 m/s, CO_2_ boils at ambient temperatures of 35 °C and 40 °C, respectively. In addition, the starting positions of two-phase flow are at 2367 m and 2175 m, respectively (i.e., 1967 m and 1775 m in the 2000 m horizontal pipeline). At the temperatures of 20 °C, 25 °C and 30 °C, CO_2_ gasification does not occur in the entire pipeline system. The depth of the vertical pipeline is also an important factor. The 200 m vertical pipelines witness the gasification of CO_2_ when the airflow velocity is 1 m/s and the ambient temperatures are in the range of 20–40 °C (Figure 6a). In contrast, at the depth of 600 m in the vertical pipeline, only part of CO_2_ gasifies at the end of a horizontal pipeline near the position of 2000 m (Figure 6c). However, at the depths of 800 m or 1000 m in the vertical pipeline, CO_2_ is transported in pure liquid phase (Figure 6d,e). It can be found that with the increase in depth, the adverse effect caused by the temperature rise of liquid CO_2_ after the intersection of the vertical and horizontal pipelines can be offset. It illustrates that deeper coal mines can effectively improve the transport distance of liquid CO_2_. In addition, the influence of the airflow velocity was also studied. According to Figure 6b, Figure 7b, Figure 8b, Figure 9b, Figure 10b and Figure 11b, at the depth of 400 m, gasification occurs under all conditions. Figure 12 illustrates the gasification positions of the horizontal pipeline at different temperatures and airflow velocities at the depth of 400 m. At the same ambient temperature (in the range of 20–40 °C), the gasification position at 1 m/s airflow velocity differs from that at 7.5 m/s by 200 m, while the distance between the gasification positions at 7.5 m/s and 15 m/s airflow velocity is over 800 m. With respect to the CO_2_ transport distance under different airflow velocities, the airflow velocity of 15 m/s corresponds to a much shorter CO_2_ transport distance than 1 m/s and 7.5 m/s. When the ambient temperature is 40 °C and the airflow velocity is 15 m/s, CO_2_ transport distance in the horizontal pipeline is merely 66 m, which suggests that CO_2_ boils almost as soon as it enters the horizontal pipeline. When the ambient temperature is 40 °C and the airflow velocities are 1 m/s and 7.5 m/s, gasification occurs at the positions of 1126 m and 284 m in the underground horizontal pipeline, respectively.

In this study, the initial position of gasification is defined as the maximum transport distance so as to ensure the absorption capacity of CO_2_ and to prevent the danger during gasification. In field practice, a 2000 m underground pipeline is unnecessary for the safe transport of CO_2_ to the injection site (gob). In this study, the real transport distance should be shorter than the maximum transport distance under specified conditions. If the transport distance fails to meet the requirement of unprotected pipelines, an insulating layer can be added to ensure that the fluid temperature stays within a certain threshold by slowing down the heat exchange between the fluid inside the pipeline and the ambient environment.

### 3.2. Insulating Layer

The simulation in Section 3.1 finds that two-phase flow occurs in most conditions of a 2000 m underground horizontal pipeline. The addition of an insulating layer can raise the thermal resistance between the fluid inside the pipeline and the ambient air, thus lowering the total heat exchange efficiency and raising the distance for safe transport. In this study, to avoid two-phase flow in the underground 2000 m pipeline, the minimum critical insulation thickness at different depths, ambient temperatures and airflow velocities were determined by dichotomy. Figure 13 displays the influences of the airflow velocity, ambient temperature and depth in different environments on the minimum critical insulation thickness.

Figure 13a–c omit the insulation thickness at 200 m depth because when the insulation thickness is below 1 m, the insulating layer cannot keep CO_2_ in pure liquid phase based on calculation. In addition, considering the space, cost, post-maintenance and safety in field practice, it is impractical to adopt an over-thick insulating layer. Consequently, the addition of an insulating layer at the depth of 200 m fails to prevent the occurrence of two-phase flow in the underground 2000 m pipeline.

Figure 13a shows the critical insulation thickness and the ambient temperature at 1 m/s airflow velocity to avoid the occurrence of two-phase flow in the pipeline. In most cases, the critical thickness is 0 m (no insulating layer is needed) because no two-phase flow occurs, including all the pipelines with a depth of 800 m, 1000 m and some 600 m pipelines. At 1 m/s airflow velocity and a depth of 600 m, the critical thickness is 0.01 cm at 35 °C and 0.07 cm at 40 °C. At a depth of 400 m, the critical thickness increases gradually with the rise of the ambient temperature. According to Figure 13a, when the depth is 400 m and the ambient temperatures are 20 °C, 25 °C, 30 °C, 35 °C and 40 °C, respectively, the critical thicknesses of the insulating layer are 0.14 cm, 0.24 cm, 0.34 cm, 0.44 cm and 0.54 cm, respectively. When the ambient temperature rises by 5 °C, the insulation thickness increases by 0.10 cm, indicating that the ambient temperature is linearly correlated with the insulating layer thickness.

Two-phase flow does not occur at 800 m and 1000 m depths in Figure 13a while it occurs at all depths in Figure 13b. When other conditions are the same, the greater the depth is, the smaller the critical insulation thickness is. For example, in Figure 13b, when the ambient temperature is 40 °C, the critical insulation thicknesses at the depth of 400 m, 600 m, 800 m, and 1000 m are 0.79 cm, 0.28 cm, 0.12 cm, and 0.04 cm, respectively. However, at the depth of 200 m, the insulating layer with a thickness below 1 m fails to prevent two-phase flow, indicating that there is no linear relationship between the depth and the critical insulation thickness. It can also be found that a longer depth of the vertical pipeline can lead to a longer underground transport distance. Vertical pipelines in coal mine with sufficient depth can reduce the thickness of the insulating layer or even remove the layer for underground transport. Even if the fluid in the vertical pipeline heats up significantly by absorbing heat from the ambient environment, the pressure gain from the gravity is greater than the pressure loss caused by the temperature rise, leading to a longer underground transport distance.

Figure 13c shares the same variations with Figure 13a,b. Airflow velocities of 7.5 m/s and 15 m/s correspond to approximate thicknesses and the thickness differences under the same conditions are no more than 0.04 cm. However, their thicknesses differ much from those at 1 m/s. This indicates that the airflow velocity exerts little influence on the critical insulation thickness in the range of 7.5–15 m/s; but at the airflow velocity of 1 m/s, the temperature and the depth of the vertical pipeline both have significant effects on the results.

To study the influence law of the insulating layer on the on-way parameters of the fluid in the pipeline, environmental factors were as follows: 15 m/s airflow velocity and 40 °C ambient temperature, and the depths are 400 m and 200 m; the influence of the insulating layer with different thicknesses on the on-way temperature and pressure was obtained.

According to Figure 14a, when there is no insulating layer, the outlet temperature of the vertical pipeline is 14.76 °C at the depth of 400 m; under an insulating layer with a thickness of 0.001 m, 0.002 m, 0.003 m, 0.005 m, 0.01 m, 0.1 m and 1 m, respectively, the outlet temperatures of the vertical pipeline are 0.67 °C, −4.82 °C, −5.12 °C, −7.66 °C, −13.31 °C, −16.46 °C and −17.02 °C, respectively. Compared with the pipelines without insulating layer, these pipelines correspond to significant lower outlet temperatures. Even an insulating layer with a thickness of 0.001 m can lead to a decrease in the outlet temperature by 14.09 °C. Furthermore, as the insulation thickness increases, the outlet temperature of the vertical pipeline decreases. The temperature curves in Figure 14a and Figure 15a continue to go up after the connection point of the vertical pipeline and the underground pipeline, yet they start to fall after the gasification point. With respect to the pressure curves, they go down after the connection point and fall at a higher rate after the gasification point. This can be explained by the same reason for the characteristic points described in Section 3.1. For the gasification points in the curves, a thicker insulating layer leads to a lower heat transfer efficiency inside and outside the pipeline, a higher thermal insulation coefficient and a longer distance of the gasification point.

Figure 14 shows the influence of an insulating layer with a thickness in the range of 0–1 m on the on-way temperature and pressure at 40 °C ambient temperature, 15 m/s airflow velocity and 400 m depth. When there is no insulating layer, the gasification point is just 66 m away from the starting position of the underground pipeline. However, after an insulating layer with a thickness of 0.001 m is added, the gasification point is at 1024 m (i.e., 624 m in the horizontal pipeline), with a distance difference of 558 m compared with the pipeline without insulating layer. Under the insulation thicknesses of 0.002 m and 0.003 m, respectively, the starting positions of gasification are at 1385 m and 1663 m (i.e., 985 m and 1263 m in the horizontal pipeline). With reference to the results of the insulating layer with a thickness of 0.001 m, it is found that with the increase in the insulation thickness, the distance of the starting position of gasification rises at a falling rate. However, when the insulation thickness reaches 0.01 m, the liquid CO_2_ in the 2000 m underground pipeline does not boil. When the insulation thicknesses are 0.01 m and 1 m, the inlet temperatures of the underground horizontal pipeline are −13.31 °C and −17.02 °C; the outlet temperatures are −0.64 °C and −15.16 °C, increasing by 12.67 °C and 1.86 °C, respectively. This phenomenon indicates that the heat transfer rate decreases with the increase in the insulation thickness; therefore, when the insulation thickness is in the range of 0.01–1 m, the temperature curves vary in a small range and tend to be horizontally straight lines. The on-way pressure experiences a much more remarkable decline after the gasification point. For pressure curves without gasification, they tend to decline linearly.

Figure 15 shows the influence of the insulating layer thickness on the on-way temperature and pressure at 40 °C ambient temperature, 15 m/s airflow velocity and 200 m depth. For pipelines without insulating layer, the gasification point is only 26 m away from the starting position of the underground horizontal pipeline, which is 40 m shorter compared with that at the depth of 400 m. The addition of an insulating layer can greatly raise the maximum safe transport distance. Even a 0.001 m thick insulating layer can raise the maximum safe transport distance from 26 m to 305 m. At the depth of 200 m, two-phase flow occurs in all simulated situations, including the pipelines with a 0.1-m-thick or 1-m-thick insulating layer. When the thickness is 0.1 m and 1 m respectively, the maximum distances jump to 1719 m and 1880 m respectively. Regardless of the practical significance of adding an insulating layer, when the insulation thickness varies from 0.01 to 1 m, the maximum transport distance merely rises by 161 m, shorter than the distance rise when the thickness varies from 0 m to 0.01 m. With the increase in the insulation thickness, the increase amount of maximum transport distance may gradually decrease. Under extreme conditions, the addition of an insulating layer fails to prevent two-phase flow.

## 4. Conclusions

In this study, the influences of different factors in the liquid CO_2_ fire-prevention pipeline system of the coal mine on the on-way parameters are studied with the aid of ASPEN HYSYS V8.4^®^ simulation software. The temperature and pressure variation characteristics of CO_2_ at different altitude differences, ambient temperatures and airflow velocities are obtained. Based on the occurrence of the two-phase flow of liquid CO_2_ during transport, this study explores whether CO_2_ in the entire transport pipeline can remain in liquid phase by adding an insulating layer and calculated the critical insulation thickness at the depths of 400 m, 600 m, 800 m and 1000 m, respectively. It is found that the pressure of gravity in the vertical pipeline can keep CO_2_ in pure liquid phase. The critical insulation thickness cannot be obtained at the depth of 200 m due to the insufficient inlet pressure from gravity. Moreover, the variations of the on-way parameters with different insulation thicknesses are also explored. This paper provides a theoretical basis for the design of a CO_2_ direct injection pipeline system, which is of great significance for the prevention and control of mine fire and the reduction of the greenhouse effect.

## Figures and Tables

**Figure 1 ijerph-19-14795-f001:**
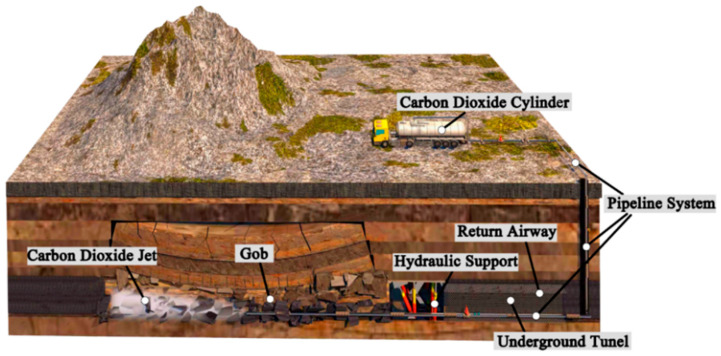
Schematic diagram of the CO_2_ pipeline transport (direct injection) system in coal mine.

**Figure 2 ijerph-19-14795-f002:**
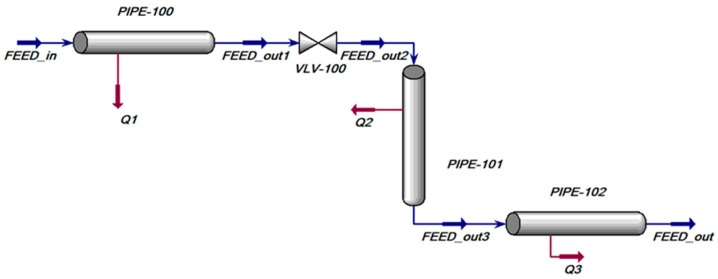
Simulation environment.

**Figure 3 ijerph-19-14795-f003:**
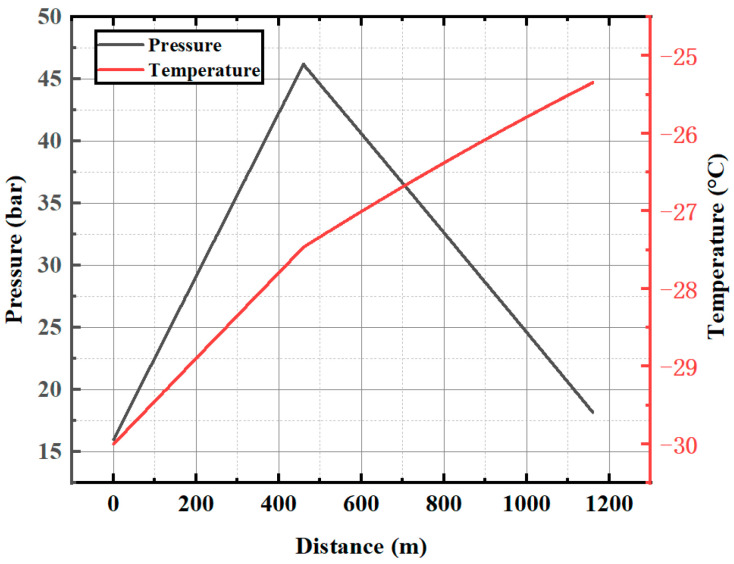
Pipeline transport simulation parameters of II020210 fully mechanized caving face in Yangchangwan Coal Mine.

**Figure 4 ijerph-19-14795-f004:**
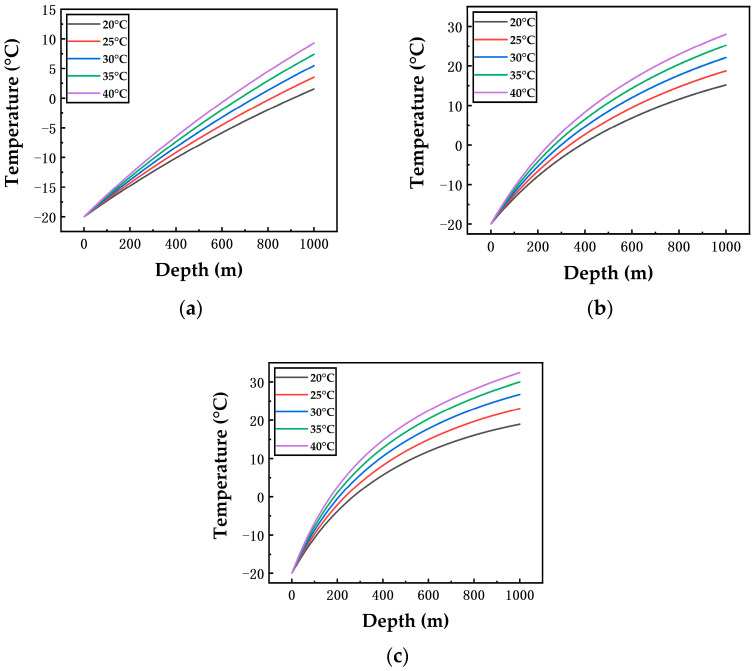
Variation of the on-way temperature with the depth in the vertical pipeline at different ambient temperatures and airflow velocities ((**a**): 1 m/s; (**b**): 7.5 m/s; (**c**): 15 m/s).

**Figure 5 ijerph-19-14795-f005:**
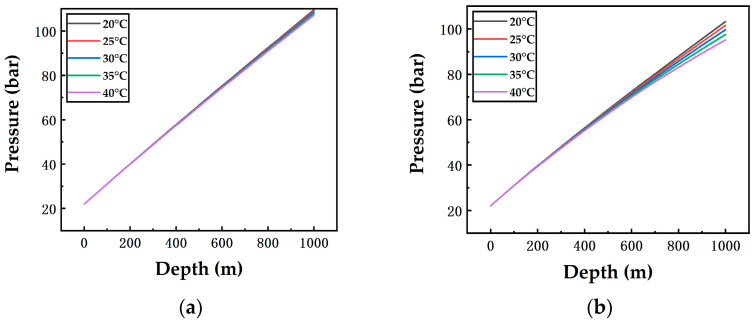
Variation of the on-way pressure with the depth in the vertical pipeline at different ambient temperatures and airflow velocities ((**a**): 1 m/s; (**b**): 7.5 m/s; (**c**): 15 m/s).

**Figure 6 ijerph-19-14795-f006:**
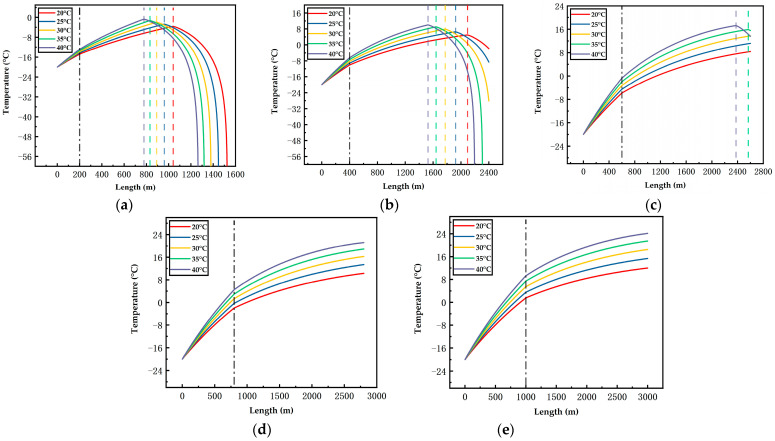
Variation of the on-way temperature with the length at 1 m/s airflow velocity, different ambient temperatures and depths ((**a**) 200 m; (**b**) 400 m; (**c**) 600 m; (**d**) 800 m; (**e**) 1000 m). The black dotted line indicates the connection between the vertical pipe and the underground pipe; The dotted lines in other colors represent gasification points.

**Figure 7 ijerph-19-14795-f007:**
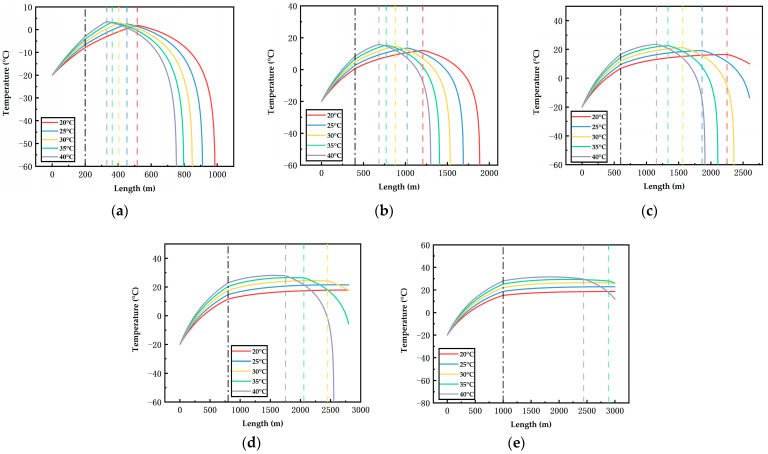
Variation of the on-way temperature with the length at 7.5 m/s airflow velocity, different ambient temperatures and depths ((**a**) 200 m; (**b**) 400 m; (**c**) 600 m; (**d**) 800 m; (**e**) 1000 m). The black dotted line indicates the connection between the vertical pipe and the underground pipe; The dotted lines in other colors represent gasification points.

**Figure 8 ijerph-19-14795-f008:**
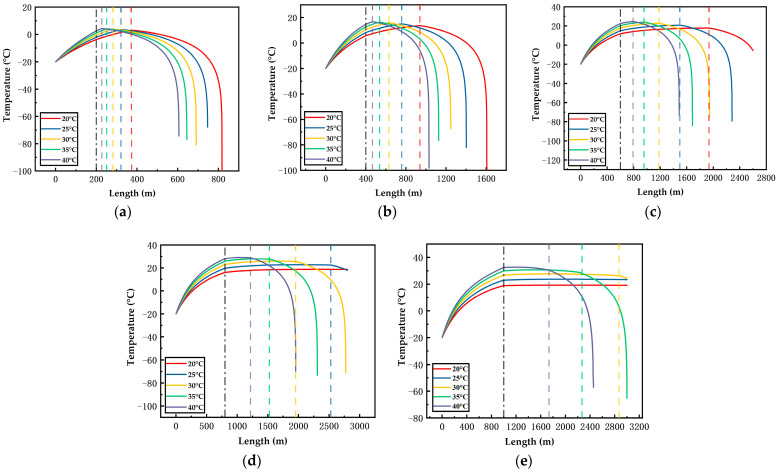
Variation of the on-way temperature with the length at 15 m/s airflow velocity, different ambient temperatures and depths ((**a**) 200 m; (**b**) 400 m; (**c**) 600 m; (**d**) 800 m; (**e**) 1000 m). The black dotted line indicates the connection between the vertical pipe and the underground pipe; The dotted lines in other colors represent gasification points.

**Figure 9 ijerph-19-14795-f009:**
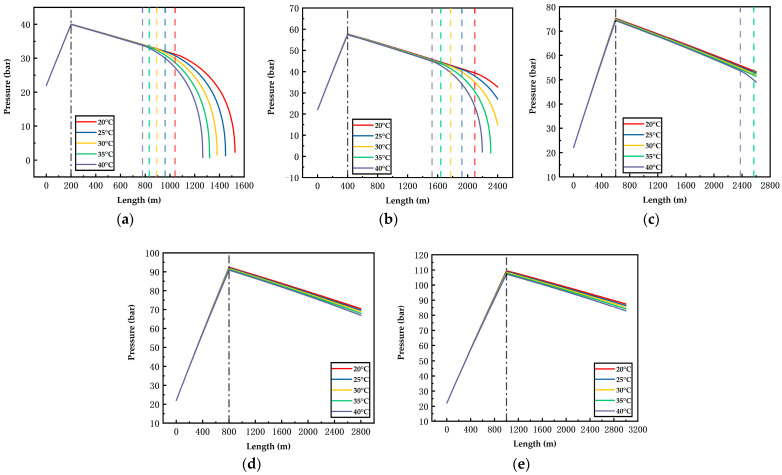
Variation of the on-way pressure with the length at 1 m/s airflow velocity, different ambient temperatures and depths ((**a**) 200 m; (**b**) 400 m; (**c**) 600 m; (**d**) 800 m; (**e**) 1000 m). The black dotted line indicates the connection between the vertical pipe and the underground pipe; The dotted lines in other colors represent gasification points.

**Figure 10 ijerph-19-14795-f010:**
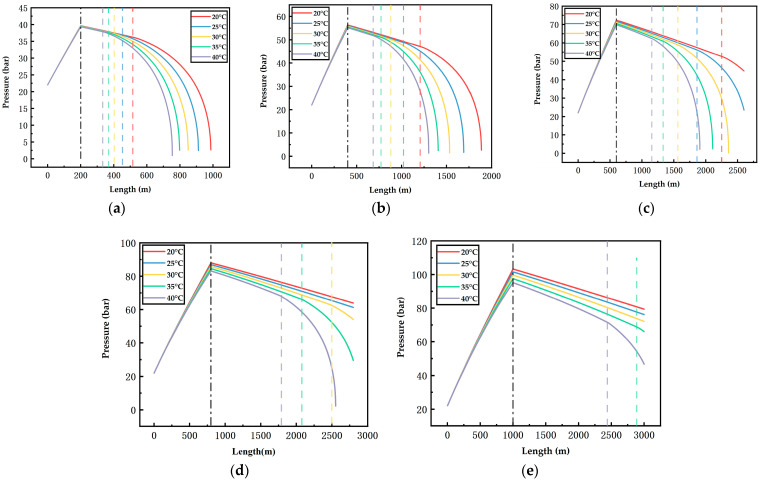
Variation of the on-way pressure with the length at 7.5 m/s airflow velocity, different ambient temperatures and depths ((**a**) 200 m; (**b**) 400 m; (**c**) 600 m; (**d**) 800 m; (**e**) 1000 m). The black dotted line indicates the connection between the vertical pipe and the underground pipe; The dotted lines in other colors indicate gasification points.

**Figure 11 ijerph-19-14795-f011:**
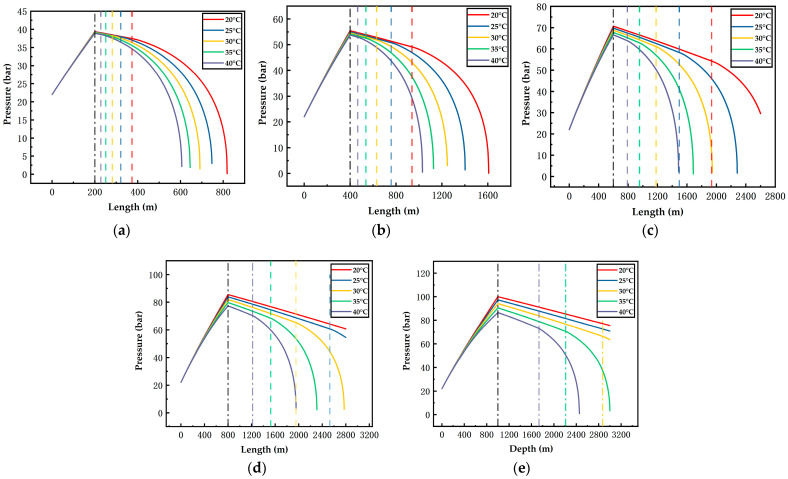
Variation of the on-way pressure with the length at 15 m/s airflow velocity, different ambient temperatures and depths ((**a**) 200 m; (**b**) 400 m; (**c**) 600 m; (**d**) 800 m; (**e**) 1000 m). The black dotted line indicates the connection between the vertical pipe and the underground pipe; The dotted lines in other colors indicate gasification points.

**Figure 12 ijerph-19-14795-f012:**
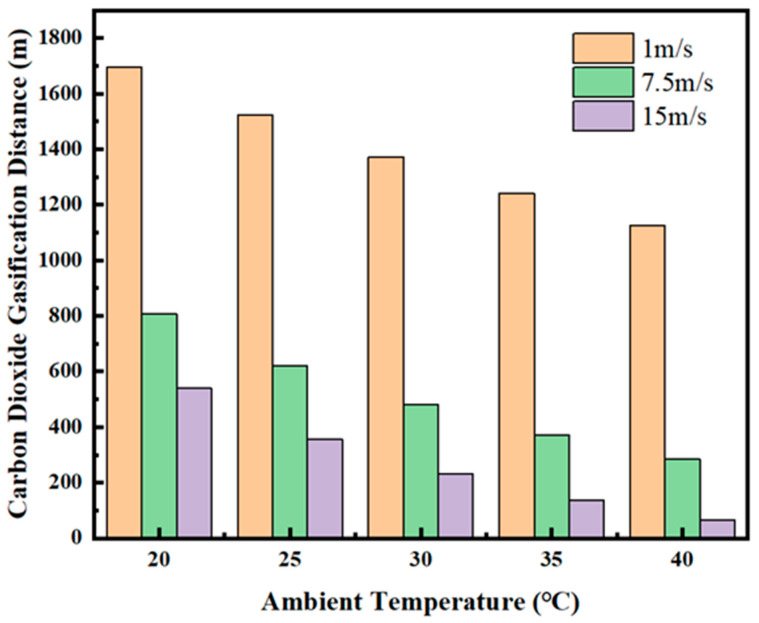
Gasification positions of different temperatures under different airflow velocities at the depth of 400 m.

**Figure 13 ijerph-19-14795-f013:**
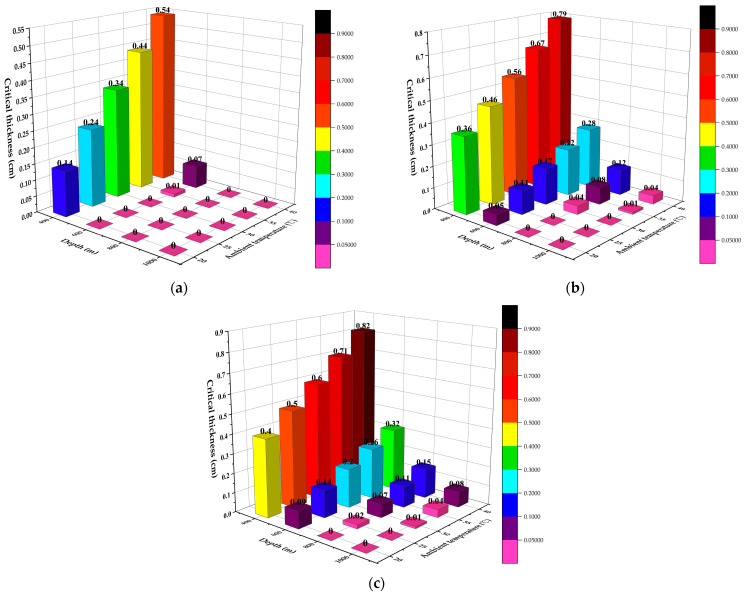
Influences of the depth and the ambient temperature on the critical insulation thickness at different airflow velocities ((**a**) 1 m/s; (**b**) 7.5 m/s; (**c**) 15 m/s).

**Figure 14 ijerph-19-14795-f014:**
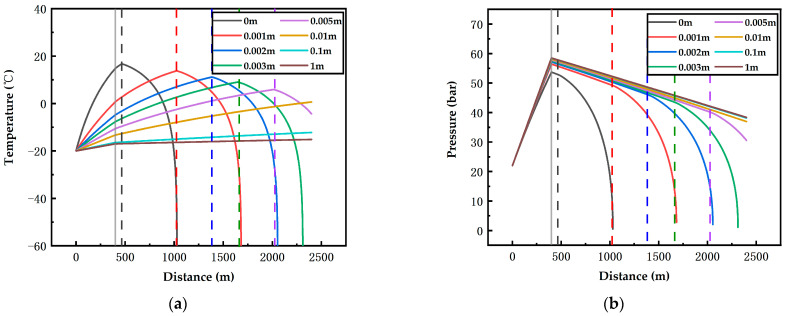
Effects of pipelines with different insulation thicknesses on the temperature (**a**) and pressure (**b**) of CO_2_ along the pipeline at 15 m/s airflow velocity, 40 °C ambient temperature and 400 m depth (grey solid line: the connection position of the vertical pipeline and the underground horizontal pipeline; colorful dotted lines: the gasification starting positions).

**Figure 15 ijerph-19-14795-f015:**
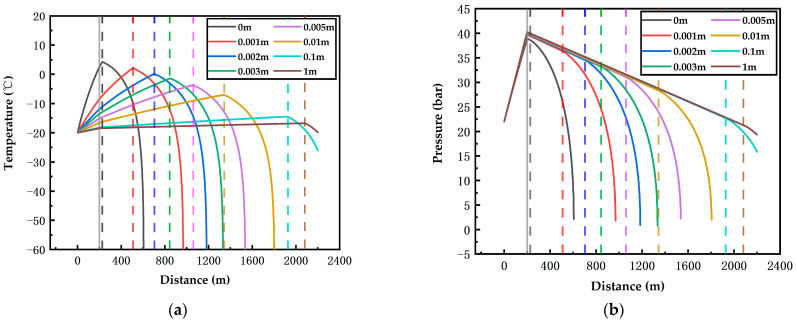
Effects of pipelines with different insulation thicknesses on the temperature (**a**) and pressure (**b**) of CO_2_ along the pipeline at 15 m/s airflow velocity, 40 °C ambient temperature and 200 m depth (Grey solid line: the connection position of the vertical pipeline and the underground horizontal pipeline; colorful dotted lines: the gasification starting positions).

**Table 1 ijerph-19-14795-t001:** Data of II020210 fully mechanized caving face in 2# coal seam of Yangchangwan Coal Mine.

Parameters	Actual Measured Values
Inlet temperature	−30 °C
Inlet pressure	In the range of 16 to 20 bar
Outlet temperature	In the range of −30 to −20 °C
Outlet pressure	In the range of 15 to 20 bar

## Data Availability

Not applicable.

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
