# Peer review of "Influence of Mine Environmental Factors on the Liquid CO2 Pipeline Transport System with Great Altitude Difference"

_ijerph, 2022, doi:10.3390/ijerph192214795_

Round 1

Reviewer 1 Report

The results of simulations of liquid CO2 flow in a mine pipeline system are presented. Using a sufficiently validated computer program, the effect of environmental factors on the possibility of transporting liquid CO2 to a gob in which there is a danger of coal spontaneous combustion was analyzed. The research procedure is presented in detail. The analysis of the results of computer simulations has enabled the Authors to formulate a number of valuable, practical conclusions on the principles of designing safe pipeline transport of liquid CO2 into underground coalmine workings.

I would suggest that the Authors expose in the Conclusions chapter the more important practical conclusions formulated in Chapter 3.

Detailed remarks

I suggest replacing the term "wind speed" with "airflow velocity" in the entire text of the article.

Line 45-46. The sentence “Carbon capture and storage (CCS), a potential and scalable technology, can reduce the amount of CO2 by 19%” is not made clear. What size represents 100%?

Line 46. Instead of: (Pires et. al.) [12, 13]. should be: [12, 13].

Lines 122, 124, 127, 128, 130. Instead of: “: the return air shaft” should be “the upcast shaft”.

Reviewer 2 Report

The authors investigated the influence of many factors on the pipeline transportation of liquid CO2. The research is interesting, the paper structure is reasonable, the English is written well, and the workload is full. I think it can be employed after modification. Some problems are as follows:

1. In lines 60, in this sentence because CO2 in other phases (supercritical phase and solid phase) ‘, what is supercritical phase? Isn't CO2 only solid, liquid, or gas three-phase

2. In lines 78, WITKOWSKI should be revised to Witkowski  .

3. In lines 172, the symbol v is inconsistent with the symbol in equation of lines 171, please recheck it.

4.  All equations in the text are not numbered, please revise it.

5. The symbols in many formulas in the text are in italics and orthography. In academic paper writing, we should strictly distinguish and use orthography and italics. Please carefully check all the symbols in the text and make changes.

6. The sentence at line 242 should be combined with line 246. A sentence is separated into paragraphs, which looks like a report rather than an academic paper.
